# Spatially Resolved Molecular Approaches for the Characterisation of Non-Invasive Follicular Tumours with Papillary-like Features (NIFTPs)

**DOI:** 10.3390/ijms24032567

**Published:** 2023-01-29

**Authors:** Isabella Piga, Vincenzo L’Imperio, Lucrezia Principi, Claudio Bellevicine, Nicola Fusco, Fausto Maffini, Konstantinos Venetis, Mariia Ivanova, Davide Seminati, Gabriele Casati, Lisa Pagani, Stefania Galimberti, Giulia Capitoli, Mattia Garancini, Andrea-Valer Gatti, Fulvio Magni, Fabio Pagni

**Affiliations:** 1Clinical Proteomics and Metabolomics Unit, Department of Medicine and Surgery, University of Milano—Bicocca, 20900 Monza, Italy; 2Department of Medicine and Surgery, Pathology, University of Milan-Bicocca, IRCCS Fondazione San Gerardo dei Tintori, 20900 Monza, Italy; 3Department of Public Health, University of Naples Federico II, 80131 Naples, Italy; 4Division of Pathology, IEO, European Institute of Oncology IRCCS, 20141 Milan, Italy; 5Department of Oncology and Hemato-Oncology, University of Milan, 20122 Milan, Italy; 6Bicocca Bioinformatics Biostatistics and Bioimaging B4 Center, Department of Medicine and Surgery, University of Milan—Bicocca (UNIMIB), 20900 Monza, Italy; 7HPB and Gastroenterological Surgery Unit, Department of Surgery, IRCCS Fondazione San Gerardo dei Tintori, 20900 Monza, Italy

**Keywords:** thyroid cancer, NIFTP, MALDI–MSI, proteomics, NGS

## Abstract

Noninvasive follicular thyroid neoplasms with papillary-like nuclear features (NIFTP) are low-risk thyroid lesions most often characterised by RAS-type mutations. The histological diagnosis may be challenging, and even immunohistochemistry and molecular approaches have not yet provided conclusive solutions. This study characterises a set of NIFTPs by Matrix-Assisted Laser Desorption/Ionisation (MALDI)–Mass Spectrometry Imaging (MSI) to highlight the proteomic signatures capable of overcoming histological challenges. Archived formalin-fixed paraffin-embedded samples from 10 NIFTPs (*n* = 6 RAS-mutated and *n* = 4 RAS-wild type) were trypsin-digested and analysed by MALDI–MSI, comparing their profiles to normal tissue and synchronous benign nodules. This allowed the definition of a four-peptide signature able to distinguish RAS-mutant from wild-type cases, the latter showing proteomic similarities to hyperplastic nodules. Moreover, among the differentially expressed signals, Peptidylprolyl Isomerase A (PPIA, 1505.8 *m/z*), which has already demonstrated a role in the development of cancer, was found overexpressed in NIFTP RAS-mutated nodules compared to wild-type lesions. These results underlined that high-throughput proteomic approaches may add a further level of biological comprehension for NIFTPs. In the future, thanks to the powerful single-cell detail achieved by new instruments, the complementary NGS–MALDI imaging sequence might be the correct methodological approach to confirm that the current NIFTP definition encompasses heterogeneous lesions that must be further characterised.

## 1. Introduction

Follicular-patterned thyroid nodules are common and include a broad range of lesions, from benign (hyperplastic nodules, adenomatoid nodules, follicular adenomas (FA)) to malignant neoplasms (follicular carcinoma (FC), follicular variant of papillary thyroid carcinoma (FVPTC)). The histopathological interpretation is usually straightforward; however, the diagnosis of a subset of cases remains an unsolved challenge, being the lesions borderline/indeterminate for malignancy. “Noninvasive follicular thyroid neoplasm with papillary-like nuclear features” (NIFTP) is the new and more accurate terminology to define very low-risk, indolent thyroid lesions, previously known as noninvasive encapsulated FV-PTC [1]. Genomics and proteomics are promising approaches to investigate the thyroid cancer landscape and to provide its molecular snapshots [2,3,4], whilst benign as well as NIFTPs are still poorly investigated. The majority of NIFTPs are characterised by the presence of RAS mutations [5], although this molecular signature is not always present, rendering the diagnosis of RAS wild-type lesions challenging, especially in the pre-operative/cytological setting. There is an urgent need to identify new possible diagnostic targets in order to support pathologists in these borderline cases. Spatial proteomics, using Matrix-Assisted Laser Desorption/Ionization (MALDI)–Mass Spectrometry Imaging (MSI), represents a cutting-edge technology to detect, directly in situ, small cell subpopulations based on their different molecular profiles, allowing the integration of molecular and histological information on tissue samples to point out early molecular alterations, even within regions that are indistinguishable at the microscopic level. MALDI–MSI proteomics has already proven its potential for the characterisation of thyroid malignancy [4], as well as its putative role as a diagnostic tool in clinical practice [6]. Some authors have already applied the MALDI–MSI approach for the study of NIFTPs [7], unveiling its discriminatory potential in this setting and opening the door for a morpho-molecular approach to this interesting disease. Moreover, our group has recently demonstrated the power of MALDI–MSI to highlight different proteomic profiles of fine-needle aspiration (FNA) biopsy cytological samples collected from NIFTPs [8]. The aim of the present study is to investigate the molecular differences of NIFTPs using a multi-omic (next generation sequencing (NGS) and MALDI–MSI) approach throughout a global characterisation of NIFTP lesions while exploring whether RAS-mutant vs. wild-type cases can be distinguished through a proteo-genetic signature.

## 2. Results

### 2.1. Clinical and Pathological Characteristics of the Cohort

The clinical, laboratory, pathological and molecular characteristics of the nine patients involved in the study (one patient contributed with two NIFTP nodules) are reported in Table 1. The mean age of the patients was 54 ± 7.21 years, the majority (78%) were females with almost all the patients having normal thyroid function. Only one case (#8) was characterised by a multinodular goitre with positive anti-thyroperoxidase (anti-TPO) antibodies (172.9 IU/mL), but still normal thyroid function (4.4 µIU/mL of thyroid-stimulating hormone, TSH). Three cases were characterised by a single nodule, the remaining having at least two bilateral nodules, three of which consisted of concurrent/incidental papillary thyroid carcinoma (PTC) (two conventional variants and one Hürthle cell variant). The mean (+/− standard deviation) diameter of the NIFTP nodule was 3.4 (± 3.21) cm, with the right and left lobes equally involved. All the nodules showed underlying genetic alterations, as per the NGS analysis, six of which (60%) had a RAS mutation, four involving the NRAS Q61R codon and two the HRAS (1 A59T and 1 Q61K).

### 2.2. Detection and Characterization of NIFTPs with MALDI–MSI

An unsupervised analysis highlighted the presence of a nodular structure in blind from the annotation performed by the thyroid pathologists on the relative H&E stained MSI (Figure 1a,b) and detected the NIFTPs (red-coloured regions) from the surrounding thyroid parenchyma (green). Most importantly, in the setting of multinodular diseases (see case #3), the MALDI–MSI distinguished the NIFTP (red) from the hyperplastic (HP) nodules (blue) with 292090 spectra Vs 146347 spectra from the HP region (blue, Figure 1c). Similar results came from the PCA, where the first three components explained 74% of the total data variability (PC1: 46%; PC2: 21%; PC3: 7%), and the score chart of the spectra of the three K-Means groups identified from the segmentation analysis (NIFTP, HP, surrounding non-nodular thyroid parenchyma (THYP), highlighted three well-separated clusters (Figure 1d).

The ROC analysis for defining a proteomic NIFTP signature found 41 signals differentially expressed between the NIFTP and HP, and 112 signals between the HP and THYP or between the NIFTP and THYP (Appendix A). Combining the MALDI–MSI and nLC–ESI-MS/MS results, 29 signals were uniquely putatively identified (Appendix A). In particular, five proteins were altered in the comparison of the NIFTP vs. the HP, with RNA Binding Motif Protein X-Linked (*m/z* 815.39, RBMX), Peroxiredoxin 1 (*m/z* 894.4, PRDX1) and Peptidylprolyl Isomerase A/Cyclophilin A (*m/z* 1505.8, PPIA) over-expressed in the NIFTPs (Appendix A and Figure 2), while Splicing Factor Proline And Glutamine Rich (*m/z* 952.35, SFPQ) and Heterogeneous Nuclear Ribonucleoprotein M (*m/z* 1063.57, HNRNPM) were both down-expressed (Appendix A). Among them, three proteins, PRDX1, SFPQ and HNRNPM, were differentially expressed in all the comparisons, being up-regulated in the nodular regions compared to the surrounding parenchyma (HP/THYP and NIFTP/THYP). A total of 23 proteins were commonly varied in the comparisons HP/THYP and NIFTP/THYP, with five peptides from collagen proteins (COL1A1, COL1A2, COL6A1, COL6A3) being always overexpressed in the surrounding parenchyma when compared to both nodular regions (Appendix A).

### 2.3. Automatic Segmentation and Proteogenomic Classification of NIFTPs

The complete list of MALDI–MSI signals (*n* = 425) was reduced to a shorter list of 10 discriminatory uniquely identified peptides among the different tissue regions (Table 2) and used for the subsequent steps of automatic segmentation by a Bisecting K-Means approach (NIFTP vs. HP vs. THYP, Figure 3), thus highlighting a specific proteomic signature able to stratify the NIFTP, HP and surrounding thyroid parenchyma regions. The NGS outputs have been used to further enrich the MALDI-based molecular classification of the NIFTPs, dividing them into RAS-mutant (#1–5 and #9) and wild-type (#6–8 and #10) (Table 1).

To obtain a combined proteogenomic classification of these tumours, the analysis of the interactome related to the RAS proteins demonstrated the presence of 561 and 896 interactions and 526 and 559 interactors for NRAS and HRAS, respectively. By LC–ESI-MS/MS, 20 out of 727 identified proteins were common interactors with NRAS and HRAS (Figure 4A). Among these, only four proteins (PPIA, Sodium/potassium-transporting ATPase subunit alpha-1 (ATP1A1), calnexin (CANX) and B Cell Receptor Associated Protein 31 (BCAP31)) for a total of seven peptides were found in our MALDI–MSI analysis and identified with an error lower than 100 ppm (Figure 4B). Two of these peptides, 1505.8 and 1584.9 *m/z,* were uniquely identified as PPIA and ATP1A1, respectively. Hence, the final list of signals that was used for the automatic segmentation included one signal for each protein (944.5, 1002.4, 1505.8, 1584.9 *m/z*). The MALDI–MSI pixel-by-pixel automatic segmentation on the NIFTP ROIs, annotated by the pathologists, showed that the NIFTPs were further stratified in two groups, which can be distinguished from the red (1–5#) or blue colour (6–8# and 10#) of the K-Means analysis (Figure 4C(a)). Interestingly, these two groups corresponded to the RAS-mutant and wild-type cases, respectively. Moreover, one of the most complex cases (#9) with an HRAS mutation demonstrated a composing blue/red proteomic landscape, suggesting both RAS-related signatures in a molecular background similar to the contralateral RAS wild-type nodule (#10).

### 2.4. Heterogeneity of RAS-Mutant NIFTPs

MALDI–MSI further unveiled the complexity of the molecular landscape of the series (Figure 4C(b)). In a multinodular background (#3), MALDI showed proteomic similarities of the HP and RAS wild-type NIFTPs (6,8# and 10#, Figure 4C). The RAS-mutant cases demonstrated a selective higher intensity of the PPIA (AUC 0.77, Figure 2). Moreover, the same protein shows no significant differences when RAS-wild-type ROIs vs. the HP region were compared (AUC 0.55, Figure 2). In a RAS-mutant NIFTP (case #5), a peripheral rim with different proteomics content and a lower presence of the PPIA peptide (Figure 5a) recapitulated a divergent histological aspect with a more microfollicular and papillary-like nucleus of the central zone of the nodule (Figure 5b, red square) as compared to a more macrofollicular aspect of the periphery (Figure 5b, blue square). Although these proteomics and morphological differences were partly sustained by the fainter staining with the NRAS Q61R immunohistochemistry in the outer (Figure 5c, blue) as compared to the inner (Figure 5c, red) part of the nodule, the laser-capture microdissection with a subsequent rtPCR analysis demonstrated the same NRAS mutation in both regions.

## 3. Discussion

The recent introduction of the NIFTP nomenclature in the WHO classification of tumours of the endocrine organs [9] downgraded a subset of carcinomas with low malignant potential (FVPTC), promising a reduction of total thyroidectomies [10,11]. Although this has been sustained by the description of a distinct molecular landscape [12], the actual definition of NIFTP still mainly relies on a histological assessment [1]. Previous attempts have been made to translate the morphological, mainly nuclear, characteristics of NIFTPs to cytological samples [13] for screening patients with suspect nodules, with different still-perfectible performances obtained in the pre-operative setting. Moreover, the recent introduction of ancillary genetic tests focused on detecting the most frequent mutations in these follicular-patterned neoplasms (e.g., RAS family) [5,14,15] is a promising adjunct to the cytological assessment of these cases. However, a substantial lack of knowledge about RAS wild-type NIFTPs still exists, and alternative molecular approaches that can complement genetics can be the way to shed light on these neglected cases. One alternative approach, based on miRNA analysis, has already shown similarities among RAS wild-type NIFTPs and follicular adenomas, suggesting a shared pathogenetic development [16]. In this setting, the application of MALDI–MSI can add further information to this puzzle, showing the differences between NIFTPs and invasive FVPTC [7]. The detection of such differences, especially on archival FFPE surgical material, through innovative, high-throughput proteomics techniques (e.g., MALDI–MSI), can represent a step forward in the identification of putative biomarkers to stratify suspect thyroid nodules in the preoperative/FNA setting, helping in the reduction of “thyroid carnage”. In this direction, the previous experience from our group already demonstrated a specific molecular signature for NIFTPs extracted from FNA samples [8]. In the present work, MALDI–MSI was able to distinguish them from surrounding normal/hyperplastic thyroid parenchyma. The NGS analysis highlighted two NIFTPs groups as RAS-mutated and RAS wild-type, and the MALDI–MSI proteomic analysis went even further in the characterisation of these challenging lesions, highlighting that RAS-mutated nodules have different signatures from RAS wild-type, and that the latter share similarities with hyperplastic lesions (Figure 4C). Indeed, our proteomic MALDI–MSI results may represent an added value in the comprehension of NIFTPs, leading us to speculate that the phenotype associated with RAS mutation could be more aggressive compared to the wild-type counterpart. This is in line with the report by Denaro et al., who demonstrated differences in the expression profiles of the miRNA of RAS-mutated and RAS-wild type, while underlining the similarities between RAS-mutated and invasive FvPTC lesions, and between RAS wild-type and follicular adenoma [16]. Conversely, one of the peptides that mainly characterised the RAS-mutant NIFTPs was Peptidylprolyl isomerase A (PPIA), also known as Cyclophilin A (CyPA), a ubiquitously distributed protein belonging to the immunophilin family, which is involved in the regulation of protein folding and trafficking with significant implications in different human diseases, including cancer [17]. In detail, the upregulation of this protein has already been linked with tumour progression and the development of metastasis [18,19], with a worse prognosis in cases of its overexpression (e.g., hepatocellular carcinoma, HCC [20]). Proteins belonging to the same PPIA family (e.g., Prolyl isomerase, Pin1) have been previously demonstrated to be overexpressed in PTC as compared to other thyroid tumours/conditions (medullary thyroid carcinoma, follicular adenoma and goitre [21]), with a subsequent demonstration of a trend in PIN1 expression towards advanced stages of PTC [22]. The recent interest in targeting the RAS pathway for cancer modulation has revealed a strong interplay of the RAS proteins with PPIA. In particular, the introduction of small molecules/drugs can perturb the GTP activity of RAS by eliciting the formation of ternary complexes with cyclophilin A [23]. This can have practical repercussions, with the introduction of tri-complex inhibitors of the oncogenic GTP-bound of RAS that can potentially overcome resistance, driven by enhanced upstream signalling [24]. Thus, cyclophilin A can represent either an adjunctive marker for RAS-mutant NIFTPs, especially in the preoperative settings as a cost-effective IHC marker for FNA triage, and can support the employment of future targeted therapies in RAS-mutant metastatic follicular-patterned thyroid neoplasms.

Other open challenges in the study of NIFTPs are centred on the histological (sprinkling effect) and molecular heterogeneity of these cases, and on the multiple/bilateral nodules scenario in the same patient, to which MALDI–MSI can be applied to provide answers. In the present work, the investigation of a unique case of bilateral disease, with either RAS-mutant (#9) or wild-type (#10) lesions, revealed a shared proteomic signature for both tumours, with still-retained RAS-related hallmarks in the former (Figure 4C(a–b)). This can be at least partly explained by the shared SLX4 mutation found in both nodules in the NGS (Table 1). The presence of this mutation has been reported in the setting of Hürtle thyroid carcinomas, along with additional genetic alterations (NRAS and ATM), but less is known about its role in NIFTPs [14]. However, the presence of a shared mutation for the two bilateral nodules can suggest a partly common genetic landscape to which the RAS hint in one of the two lesions provided additional proteomics modifications.

Finally, the investigation of intranodular heterogeneity, in the present study, demonstrated the capability of MALDI to unveil subtle differences among the centre and the periphery of RAS-mutant NIFTPs, which can even recapitulate the morphological aspects detected in “conventional” light microscopy assessment (as exemplified here by case #5), even with a similar genetic signature (NRAS Q61R) found in the different regions after microdissection. In this particular case, the molecular heterogeneity unveiled by MALDI–MSI can be at least in part explained by a possible acquisition of additional genetic events (e.g., NOTCH2 mutation), which have already been reported in other forms of thyroid tumours and are thought to worsen the biological aggressiveness of the tumour [25]. Although promising and sustained by a complete clinical, histological and genetic background, these results must be supported by validation by independent cohorts enriched in further NIFTP cases with a comprehensive genetic study. This, hopefully, will help in finding additional signatures, especially for RAS wild-type cases, still orphans of a unique genetic signature, and to support the eventual translation of PPIA as a marker of RAS-mutant NIFTPs, especially in the preoperative/FNA setting.

## 4. Materials and Methods

### 4.1. Pathology

Archived formalin-fixed paraffin-embedded (FFPE) samples and the corresponding diagnostic slides of 10 NIFTPs were retrieved from the archives of the Department of Pathology, IRCCS Fondazione San Gerardo dei Tintori (University of Milano-Bicocca (UNIMIB)), Italy. All the cases were reviewed by three experienced thyroid pathologists (FP, CB, FM) to confirm the diagnosis of NIFTP, based on the morphological, immunohistochemical and molecular criteria of the WHO classification [26]. In particular, all the selected cases showed an encapsulation or clear demarcation, follicular growth pattern with no papillae, solid, trabecular or insular patterns in a total of less than 30% of the total tumour volume, and no psammoma bodies and nuclear features of PTC (enlargement, crowding/overlapping, elongation, irregular contours, grooves, pseudoinclusions, chromatin clearing with a nuclear score of 2 or 3). Approval was obtained from the local ethical committee (FINAL-TIR PU 3581/21). For the immunohistochemistry, NRAS Q61R was tested (rabbit monoclonal antibody, clone RST-NRAS, dilution 1:20) on DAKO Omnis (Agilent, Santa Clara, CA, US). Seven unstained slides with 4 μm-thick sections from representative FFPE tissue blocks were used for NGS analysis. Manual microdissection was performed before nucleic acid isolation to enrich the tumour cell content using a sterile scalpel. The DNA was extracted using the Maxwell RSC DNA FFPE Kit (Promega, Madison, WI, USA), following the manufacturer’s instructions, and then quantified by the QuantiFluor ONE dsDNA System (Promega, Madison, WI, USA) on the Quantus Fluorometer (Promega, Madison, WI, USA). The mutational analyses were performed on each nodule through the NGS panel Oncomine Comprehensive Assay (OCA) v3 System (ThermoFisher Scientific, Waltham, MA, USA), which evaluates the mutational status (single-nucleotide variants, SNV), insertions/deletions, and copy-number variations (CNV) of 161 cancer-related and clinically actionable genes. A full list of the genes included in this panel is available online (https://www.Thermofisher.com/order/catalog/product/ A35805, accessed on 1 September 2022). Briefly, 10 ng of genomic DNA was used for the library preparation, and the subsequent Ion 540 chip (ThermoFisher Scientific, Waltham, MA, USA)-loading was performed automatically on the Ion Chef System (ThermoFisher Scientific, Waltham, MA, USA). The sequencing was performed using the Ion S5 System (ThermoFisher Scientific, Waltham, MA, USA), and the data were analysed using the Ion Reporter Software (v. 5.16) (ThermoFisher Scientific, Waltham, MA, USA). Only the mutations with an allele frequency ≥5% and with adequate quality metrics were reported. The mutations were classified as actionable/pathogenic based on the annotation in three different publicly available cancer genomics data sets (i.e., cBioPortal, https://www.cbioportal.org/,36, 37; ClinVar, https://www.ncbi.nlm.nih.gov/clinvar/; and the Catalogue of Somatic Mutations in Cancer, COSMIC, https://cancer.sanger.ac.uk/cosmic, accessed on 1 September 2022). The clinically relevant and borderline alterations were visually inspected using the Integrative Genomics Viewer (IGV) software (Broad Institute, Cambridge, MA, USA). The median absolute pairwise difference (MAPD) metric was used to identify low-quality samples at risk of generating false results and therefore needing to be excluded; only the cases with MAPD of <0.5 were included. For the identification of the somatic variants, the read depth of at least 1000× was respected. Finally, the pathogenetic variants detected in the NGS were appropriately confirmed through the application of orthogonal molecular methods, such as the quantitative Polymerase Chain Reaction (qPCR) instrument, EasyPGX, with the EasyPGX ready Thyroid panel (Diatech Pharmacogenetics, Jesi, Italy), which is a targeted, multi-biomarker assay that enables detection of the principal mutations of the exon 2 (12 and 13 codons) and exon 3 (61 codon) of the KRAS, NRAS, HRAS genes, and of the 600 and 601 codons of the BRAF gene from the FFPE tumour tissue. The limits of detection of the EasyPGX ready THYROID kit are: BRAF K601E/V600_K601>E 2%; BRAF V600E/Ec 1%; KRAS G12x-G13D 5%; KRAS Q61x 2%; NRAS G12x-G13x 7%; NRAS Q61x 1%; HRAS G12x-G13R 5%; HRAS Q61x 7%. Laser-capture microdissection, using the LMD7 instrument (Leica Microsystem, Wetzlar, Germany), was performed on selected cases to investigate the intratumoral genetic heterogeneity. Laser microdissection was performed for each area to maximise the genetic content of the dissected tissue.

### 4.2. Sample Preparation for MALDI–MSI

For the MALDI–MSI, the FFPE samples were treated as previously described [27]. Four-micron-thick sections were cut and mounted onto conductive indium tin oxide (ITO) glasses. The slides were stocked at room temperature until the day of the analysis. Each slide was treated using our previously published method [27], comprising dewaxing with toluene, tissue rehydration with decreasing concentrations of ethanol and water, and, finally, citric acid antigen retrieval (10 mM citric acid buffer and heated in a water bath at 97 °C, pH 5.95 for 45 min). For the protein digestion, uniform layers of trypsin were deposited using an iMatrixSpray automated spraying system (Tardo Gmbh, Subingen, Switzerland) with an optimised method (heat-bed temperature: 37 °C; number of spray cycles: 15; enzyme density: 1.2 µL/cm^2^; movement speed: 160 mm/s; distance between spray lines: 2 mm; needle height: 45 mm), and the tissues were then incubated overnight at 40 °C in a humidity chamber. After the enzymatic digestion, a matrix deposition for the MALDI–MSI analysis was performed by spraying α-cyano-4-hydroxycinnamic acid (10 mg/μL in 70% ACN, 30% H2O and 1% TFA) using HTX TM-Sprayer (HTX Technologies, LLC, Chapel Hill, NC, USA) with the following parameters: temperature 75 °C; number of passes 4; flow rate 0.12 mL/min; velocity 1200 mm/min; track spacing 2 mm; and pressure 10 psi.

### 4.3. MALDI–MSI Analysis

The mass spectra were acquired in the reflectron positive mode, within the *m/z* 700 to 3000, using a rapifleX MALDI Tissuetyper (Bruker Daltonik GmbH, Bremen, Germany) MALDI-TOF/TOF MS equipped with a Smartbeam 3D laser operating at 2 kHz frequency. The MALDI–MS images were acquired with a single-spot laser setting of 50 μm and a scan range of 46x46 μm. A mixture of standard peptides within the mass range of *m/z* 750 to 3150 (PepMix I, Bruker Daltonik, Bremen, Germany) was used for the external calibration, directly applied on the glass slide (mass accuracy < 10 ppm). A FlexControl 4.0 (Bruker Daltonik, Bremen, Germany) was used to set up the instrument parameters for the acquisition method, and FlexImaging 5.0 (Bruker Daltonik, Bremen, Germany) for the MALDI–MSI analysis visualisation. Following the MALDI–MSI analysis, the matrix was collected from each slide with 70% ACN, 30% H_2_O and 1%TFA in order to perform the protein extraction, transferred into an Eppendorf Tube and stored at −20 °C before the nLC–ESI-MS/MS analysis. Hence, the remaining matrix was removed with increasing concentrations of ethanol (70%, 90% and 100%), and the slides were stained with haematoxylin and eosin (H&E). Finally, the slides were converted to a digital format using a MIDI II digital scanner (3DHISTECH, Budapest, Hungary), allowing the integration of the proteomic and morphological data. The regions of interest (ROIs) of the NIFTP nodules were annotated by the pathologist.

### 4.4. MALDI–MSI Data Analysis

The data files containing the individual spectra of each entire measurement region were then imported into SCiLS Lab 2023a Pro software (http://scils.de/, accessed on 1 July 2022; Bremen, Germany) to perform a pre-processing: baseline subtraction (TopHat algorithm), normalisation (Total Ion Current algorithm) and spatial denoising. The average (avg) spectra, representative of the whole measurement regions, were generated in order to display the differences in the protein profiles. Subsequently, SciLS Lab software was used to define the ROIs that were previously annotated by the pathologist. Peak picking and alignment were performed for the feature extraction for statistical analysis, and this resulted in the detection of 425 *m/z* features within the dataset. Furthermore, a Bisecting K-Means algorithm was also performed: the individual spectra showing a similar shape were grouped and uniformly coloured; hence, the pixels of the MS tissue image were coloured according to the different identified clusters. A dimensionality reduction was performed through principal component analysis (PCA) on the individual spectra in order to show the data in a 3D setting, using unit variance scaling and weak denoising. A receiver operating characteristic (ROC) curve analysis was performed, with an area under the curve (AUC) ≥ 0.75 and a *p*-value ≤ 0.001 (calculated with the Wilcoxon rank-sum test) being required for a peak to be considered statistically significant. The open-source software mMass v.5.5 (http://www.mmass.org, accessed 1 July 2022) was used for the mass spectra visualisation [28,29].

### 4.5. nLC–ESI-MS/MS Sample Preparation and Analysis

After the MALDI–MSI analysis, an nLC–ESI-MS/MS analysis for peptide identification was performed. The matrix collected from each slide was dried with a vacuum centrifugal evaporator (Hetovac, Savant) until a final elution volume of about 20 μL was achieved and resuspended in 100 µL of phase A (98:2:0.1; water/acetonitrile/trifluoroacetic acid). The nLC–ESI MS/MS analysis was performed using a Dionex UltiMate 3000 rapid separation (RS) LC nano system coupled with an Impact HD UHR-QqToF (Bruker Daltonik, Germany). The sample desalting and concentration were carried out using a pre-column (Dionex, Acclaim PepMap 100 C18, cartridge, 300 μm), and the peptides were separated with a 50 cm column (Dionex, ID 0.075 mm, Acclaim PepMap100, C18) at 40 °C, using a 120 min gradient from 96% to 2% of phase A (0.1% formic acid), whilst phase B was 0.08% formic acid:acetonitrile (80:20). The MS was operating in data-dependent acquisition mode. Compass DataAnalysis v4.1 software (Bruker Daltonics, Hamburg, Germany) was used to calibrate, deconvolute and convert the acquired raw data prior to the protein identification and quantification. The Peaks Studio X-Plus (Bioinformatics Solutions Inc., Waterloo, ON, USA) was used for protein identification. The parameters were set as follows: trypsin as the digestive enzyme, no fixed modifications, oxidation (M) and FFPE+12 and FFPE+30 as the variable modifications, the precursor mass error and the fragment mass error tolerances were set at 20 ppm and 0.05 Da, respectively. The identification engine uses the Swissprot database and the homo sapiens taxonomy was selected. For identification, an FDR ≤ 1% was applied. The proteins were considered to be identified if there was at least one unique significant peptide (*p*-value ≤ 0.05). The signals of interest present within the MALDI–MSI dataset were correlated with the positively identified peptide sequences obtained using the nLC–ESI-MS/MS. An identification was putatively assigned to a signal if an error of <100 ppm was noted between the two measured *m/z* values. The IntAct molecular interaction database open-source tool (https://www.ebi.ac.uk/intact/home, accessed on 1 July 2022) was used to find the proteins interacting with NRAS and HRAS [30,31]. The Venny 2.1 tool was used to build the Venn diagram (https://bioinfogp.cnb.csic.es/tools/venny/, accessed the 1 July 2022).

## 5. Conclusions

The combined application of genetic and proteomics techniques is promising for the investigation of NIFTPs, as demonstrated by the NGS/MALDI–MSI approach reported here. The identification of a commonly deregulated peptide (PPIA) in the RAS-mutant form of NIFTPs is a first step towards a better comprehension of this cryptic thyroid neoplasm. A larger series including very challenging multifocal and bilateral cases may further enrich our knowledge on the topic.

## Figures and Tables

**Figure 1 ijms-24-02567-f001:**
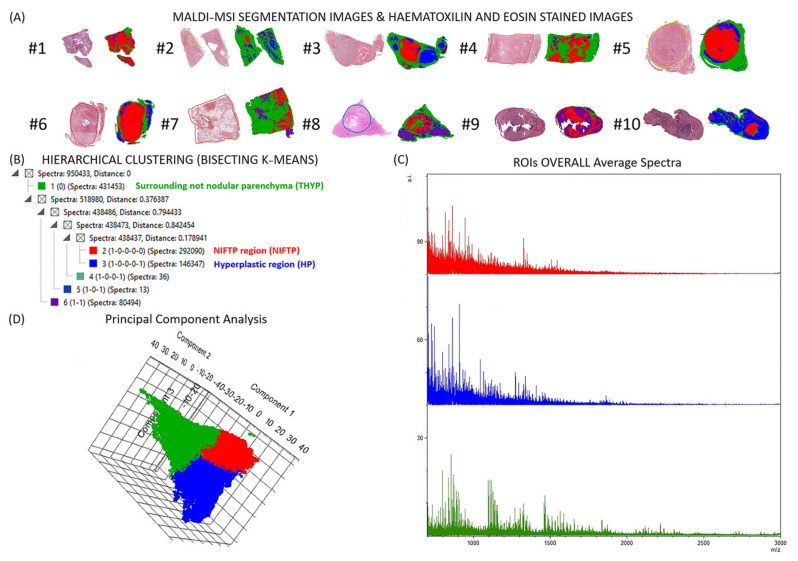
(**A**). Molecular maps obtained from MALDI–MSI compared to the annotation of NIFTPs from the expert pathologists. (**B**). Bisecting K-Means segmentation cluster tree of the tissue, based on the proteomic data, allowed the distinction of NIFTPs (red) from hyperplastic regions (HP, blue) and surrounding non-nodular thyroid parenchyma (THYP, green). (**C**). Average spectra of these three different regions. (**D**). Unit variance principal component analysis (PCA) score plot of the individual spectra.

**Figure 2 ijms-24-02567-f002:**
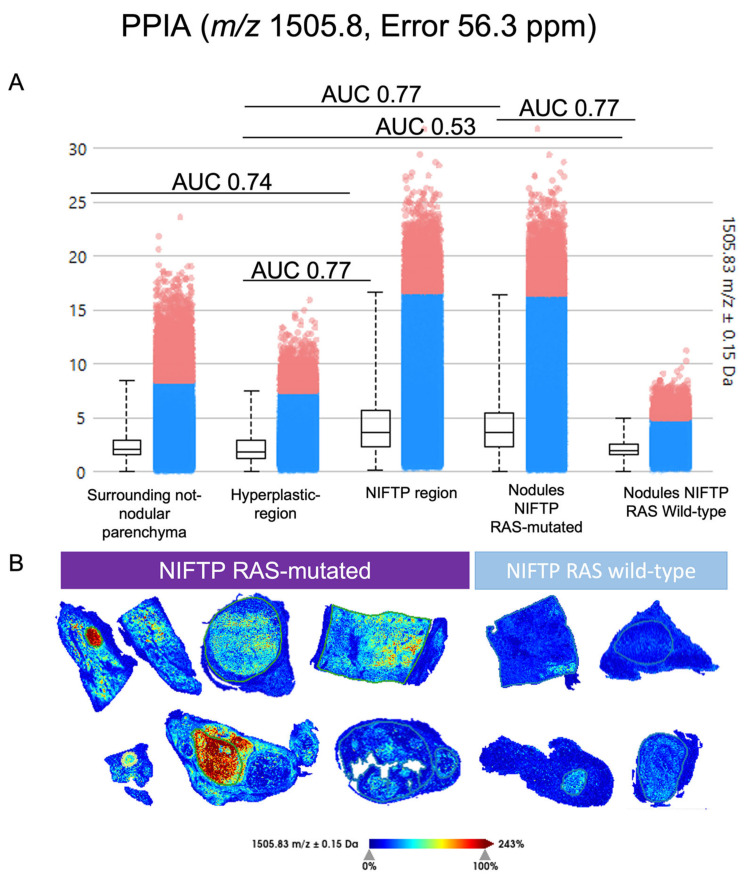
Spatial localization of Peptidylprolyl Isomerase A/Cyclophilin A (PPIA) signal *m/z* 1505.8. (**A**). Intensity box plots and AUC for all comparisons. Blue dots represent the spectra where intensities of the ion image are between the lower and upper quartiles; red dots represent the spectra whose intensities are outliers. AUC: area under the curve, and (**B**). MALDI–MSI images.

**Figure 3 ijms-24-02567-f003:**
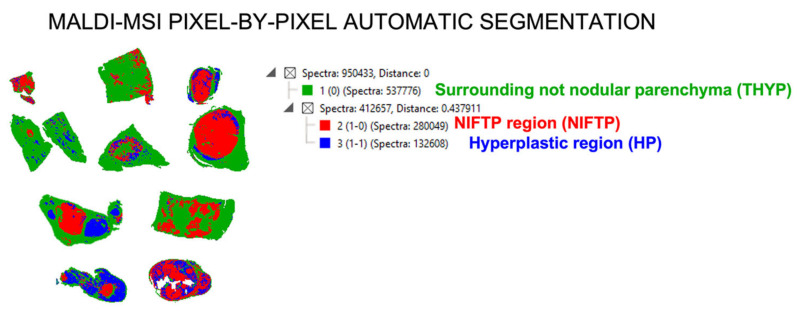
Molecular maps obtained with a K-Means approach starting from the 10-peptide signature discovered with MALDI–MSI. This demonstrates the capability of this approach to identify the NIFTPs (red areas) from the remaining structures (HP and THYP).

**Figure 4 ijms-24-02567-f004:**
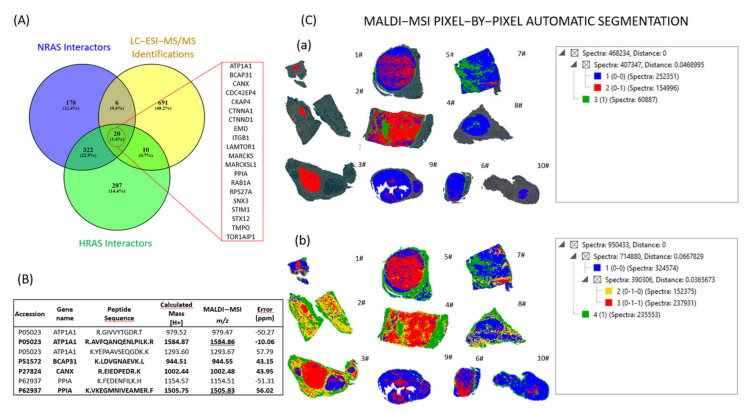
Integration of RAS protein–protein interaction analysis and LC–ESI-MS/MS and MALDI–MSI results. (**A**). Venn diagram showing the number of common proteins among the NRAS protein interactors; HRAS protein interactors and proteins putatively identified in the MALDI-matrix by LC–ESI-MS/MS. (**B**). List of proteins found in MALDI–MSI analysis and putatively identified with an error lower than 100 ppm; in bold are signals uniquely identified and all the signals used for the automatic segmentation are underlined. (**C**). MALDI–MSI-based automatic segmentation using (a) only the NIFTPs ROIs annotated by the pathologists and (b) all the tissue regions.

**Figure 5 ijms-24-02567-f005:**
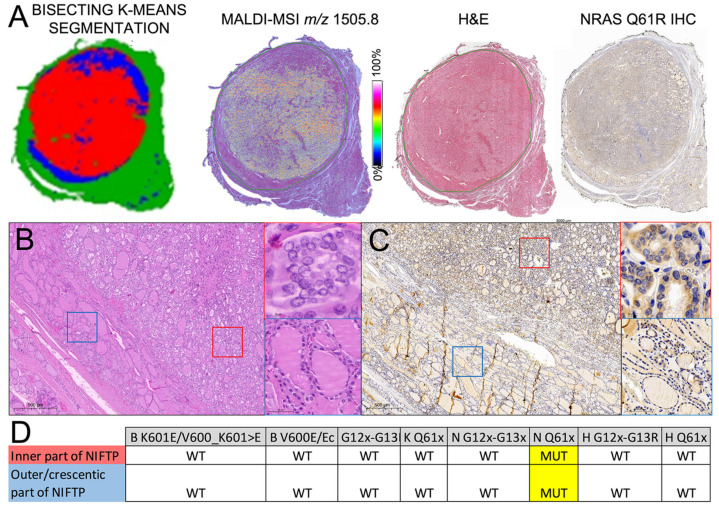
The comparative analysis of images obtained from the K-Means segmentation, MALDI–MSI (based on 1505.8 m/z, PPIA), routine H&E and NRAS Q61R IHC (**A**, from left to right) of an emblematic case (#5) demonstrates the presence of intranodular heterogeneity of this NIFTP. In particular, the molecular differences highlighted by the K-Means and MALDI–MSI between the inner part of the nodule and the outer rim are confirmed by morphology, with a more microfollicular growth and clear/overlapped papillary-like nuclei with grooves in the inner (**B**, red square) as compared to the more macrofollicular pattern without significant nuclear alterations of the outer part of the lesion, with a goitre-like appearance on H&E (B, blue square). Although the NRAS Q61R IHC shows fainter staining of the outer (**C**, blue square) as compared to the inner (**C**, red square) part of the nodule, further supporting phenotypic differences of the two regions highlighted by proteomics, the rtPCR analysis after laser-capture microdissection demonstrated an NRAS Q61x mutation in both the components without concurrent genetic abnormalities (**D**). This could be explained by intranodular heterogeneity in a single clonal NRAS-mutant follicular proliferation with overall NIFTP characteristics. See Appendix A to evaluate the whole slide image of the case.

**Table 1 ijms-24-02567-t001:** Clinical, pathological and molecular characteristics of the cohort. *Single patient with bilateral NIFTP.

Case No.	Age	Sex	No. Nodules	Diameter (cm)	Thyroid Lobe	Mutational Status		Other Thyroid Diseases	Serology		
						*RAS*	Others		TSH (µIU/mL)	anti-TPO Ab (IU/mL)	anti-Tg Ab (IU/mL)
1	65	F	3	0.3	Left	NRAS Q61R	PTCH1, RAD51B, SMARCA4	Hürthle cell carcinoma—Right lobe	1.9	8	11
2	52	F	2	0.5	Left	HRAS A59T	RICTOR, CHEK2, SLX4	Multinodular goitre and C cell hyperplasia	4.12	10	10
3	50	F	multiple	1.3	Right	NRAS Q61R	-	PTC— Right lobe	2.26	9	12
4	41	F	1	5.5	Right	NRAS Q61R	-	-	3.12	7	9
5	48	M	3	3	Right	NRAS Q61R	NOTCH2	-	0.6	8	11
6	61	F	1	1.6	Left	WT	NTRK1, FBXW7	-	1.5	10	12
7	58	F	1	9	Left	WT	MDM4, ESR1, CDKN2A, AXL	microPTC—Right lobe	1.16	9	11
8	53	F	multiple	1	Right	WT	SLX4, ATM, ARID1A, AXL	Multinodular goitre	4.4	172.9	2.83
9*	56	M	2	2	Left	HRAS Q61K	SLX4	-	0,85	5	7
10*	56	M	2	2	Right	WT	SLX4	-	0,85	5	7

**Table 2 ijms-24-02567-t002:** List of the 10 uniquely identified signals used for automatic segmentation. Over-expressed peptides are highlighted in red, under-expressed ones in green. THYP: Surrounding not nodular thyroid parenchyma, HP: Hyperplastic.

m/z	ROC Analysis	Gene Name
815.39	NIFTP/HP	RBMX
	NIFTP/THYP	
836.48	HP/THYP	COL1A1
	NIFTP/THYP	
868.39	HP/THYP	COL1A2
	NIFTP/THYP	
894.42	HP/THYP	PRDX1
	NIFTP/HP	
	NIFTP/THYP	
952.35	HP/THYP	SFPQ
	NIFTP/THYP	
	NIFTP/HP	
1063.57	HP/THYP	HNRNPM
	NIFTP/HP	
	NIFTP/THYP	
1095.62	HP/THYP	COL1A2
	NIFTP/THYP	
1320.65	HP/THYP	COL6A3
	NIFTP/THYP	
1505.83	NIFTP/HP	PPIA
	NIFTP RAS+/NIFTP RASWT	
1580.75	HP/THYP	COL6A1
	NIFTP/THYP	

## Data Availability

Data that support the findings of this study are available upon reasonable request from the corresponding author F.P.

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
