# Peer review of "Spatially Resolved Molecular Approaches for the Characterisation of Non-Invasive Follicular Tumours with Papillary-like Features (NIFTPs)"

_ijms, 2023, doi:10.3390/ijms24032567_

Round 1

Reviewer 1 Report

The authors demonstrated that PPIA was overexpressed in the RAS-mutant NIFITP using the NGS/MALDI-MSI. The methodology is cutting-edge, and the results are clear. There are some issues to be reconcile.

1) In the Fig 5 B, it is not clear if the blue part is really a NFITP or not. To me, it seems goiter like lesion. I would suggest authors to show the whole slide image of the represented slide.

2) Practically, the method seems very expensive, and only applied postoperative cases; it looks far from standard diagnostic method. Minimally invasive follicular thyroid carcinoma and NIFTP needs almost the same follow up once it underwent operation. I think we need d preoperative diagnosis, not postoperative diagnosis. What is the most feasible goal of this method?

Author Response

The authors demonstrated that PPIA was overexpressed in the RAS-mutant NIFITP using the NGS/MALDI-MSI. The methodology is cutting-edge, and the results are clear. 

We would like to thank the reviewer for appreciating our work and providing useful comments to improve the manuscript.

There are some issues to be reconcile.

Thanks, we made our best to address the comments as detailed below.

1) In the Fig 5 B, it is not clear if the blue part is really a NIFTP or not. To me, it seems goiter like lesion. I would suggest authors to show the whole slide image of the represented slide.

Thanks for pointing out this crucial point. Actually, the case represented in Figure 5 has been selected because it represents a perfect example of intranodular phenotypic heterogeneity in an otherwise clonally (NRAS-mutant) follicular proliferation with main cytological features compatible with NIFTP. In detail, the blue square cited by the reviewer represents the surrounding/peripheric crescent-shaped region of the nodule, and we agree with the reviewer that (1) on H&E this region looks more an hyperplastic component with macrofollicular growth and no significant nuclear alterations (as in the blue inset) and (2) the negativity of NRAS IHC further support the hypothesis of different phenotypic aspects of the two areas. However, the laser-capture guided genetic analysis of these two components revealed that both are NRAS Q61R mutant, sharing a common driver mutation which makes them all part of the same lesion, with prevalent NIFTP-like features. We chose this because it nicely shows how MALDI-MSI, in a unique genetically defined tumor, can demonstrate proteomic intralesional heterogeneity, pairing the morphological findings on H&E. However, we tried to better rephrase these aspects in the Figure 5 legend, as well as by adding as supplementary material the virtual slide of the case so that you and readers can review the histological aspects of this peculiar case.

2) Practically, the method seems very expensive, and only applied postoperative cases; it looks far from standard diagnostic method. Minimally invasive follicular thyroid carcinoma and NIFTP needs almost the same follow up once it underwent operation. I think we need d preoperative diagnosis, not postoperative diagnosis. What is the most feasible goal of this method?

Thanks for stressing this point and we agree with you that the most pressing clinical need is the detection of diagnostic biomarkers to detect tumors requiring surgery in the preoperative/FNA setting. In this sense, the analysis of challenging lesions (e.g. NIFTP) with high throughput proteomics techniques directly in situ on archival FFPE sections can allow the detection of candidate biomarkers that can eventually be translate on the preoperative setting for the better stratification of thyroid nodules, as would be the case for PPIA in this study. In particular, the preoperative assessment of NIFTPs can be challenging and often these cases will fall in the THY3 or AUS/FLUS cytological category, thus requiring surgery for diagnostic purposes for the final differential with follicular invasive variant of PTC. Introducing cost-effective tools (e.g. IHC) to detect biomarkers discovered with these next generation proteomics techniques can be the solution to fight the “thyroid carnage”. Finally, these techniques will allow to obtain from a single analysis of an FFPE tissue section a significant amount of molecular information in one single analysis while the analysis of multiple biomarkers using conventional IHC assays is not only time-consuming but it also requires a larger number of serial tissue sections. Even if the advent of multiplexed immunohistochemistry combined with fluorescence microscopy could overcome some of the problems associated with conventional IHC, still the cost of the antibodies necessary for the experiments might be equal or even higher than a single MALDI-MSI mass spectrometry study which will however allow to gain more comprehensive molecular information on the disease. We tried to make the clinical relevance of the findings more evident in the Discussion by adding some speculative aspects regarding the possible impacts of these discoveries on archival NIFTPs and their possible repercussions on the preoperative setting.

Reviewer 2 Report

Article entitled „Spatially Resolved Molecular Approaches for the Characterization of Non-invasive Follicular Tumours with Papillary-like Features (NIFTPs)” promotes an innovative approach NGS/MALDI-MSI to better characterize NIFTPs. In this study authors were able to identify a commonly deregulated in the RAS-mutant form of NIFTPs peptide (PPIA) and  performed comparative analysis of RAS-mutant NIFTP. This work is well designed. In my opinion the manuscript is written in an understandable way and should be published after text editing (checking the punctation).

Author Response

Article entitled „Spatially Resolved Molecular Approaches for the Characterization of Non-invasive Follicular Tumours with Papillary-like Features (NIFTPs)” promotes an innovative approach NGS/MALDI-MSI to better characterize NIFTPs. In this study authors were able to identify a commonly deregulated in the RAS-mutant form of NIFTPs peptide (PPIA) and  performed comparative analysis of RAS-mutant NIFTP. This work is well designed. In my opinion the manuscript is written in an understandable way and should be published after text editing (checking the punctation).

We would like to thank the reviewer for its positive feedback. We tried our best to address the comments of Reviewer #1, hoping this would help in improving the manuscript value, and systematically revised the text checking the punctuation as properly suggested.

Round 2

Reviewer 1 Report

The authors made appropriate revisions.